# Efficacy of Platelet-Rich Plasma in Women with a History of Embryo Transfer Failure: A Systematic Review and Meta-Analysis with Trial Sequential Analysis

**DOI:** 10.3390/bioengineering10030303

**Published:** 2023-02-27

**Authors:** Eduardo Anitua, Mikel Allende, María de la Fuente, Massimo Del Fabbro, Mohammad Hamdan Alkhraisat

**Affiliations:** 1Regenerative Medicine Department, BTI Biotechnology Institute, 01007 Vitoria-Gasteiz, Spain; 2Clinical Research, University Institute for Regenerative Medicine and Oral Implantology (UIRMI), 01007 Vitoria-Gasteiz, Spain; 3Department of Biomedical, Surgical and Dental Sciences, University of Milan, 20122 Milan, Italy; 4Fondazione IRCCS Ca’ Granda Ospedale Maggiore Policlinico, 20122 Milan, Italy

**Keywords:** repeated implantation rate, thin endometrium, platelet-rich plasma, clinical pregnancy, biochemical pregnancy

## Abstract

Assisted reproductive technology (ART) is used to enhance pregnancy in infertile women. In this technique, the eggs are removed from the ovary and fertilized and injected with sperm to make embryos. Unfortunately, embryo implantation failures still occur in many of these women. Platelet-rich plasma (PRP) therapies use a patient’s own platelets to promote tissue healing and growth, including endometrium. The growth factors provided by the platelets play a criterial role on the regenerative ability of PRP. In the last years, PRP treatments have been gaining a lot of popularity to treat women with repeated ART failures. In this study, we collected and summarized all information published in the scientific literature to assess the evidence of the PRP effect on pregnancy. We only considered randomized controlled trials (RCT), a type of study designed to be unbiased and considered at the highest level of evidence. Our analysis indicates that PRP therapies might be an effective treatment in cases of poor responsiveness to conventional ART. However, additional studies (well-designed) are necessary to confirm this beneficial effect of PRP.

## 1. Introduction

Around 10–15% of couples suffer from infertility or have difficulties conceiving [1,2]. The introduction of assisted reproductive technology (ART) has had a favorable impact on the pregnancy rate in infertile women. However, implantation failures still occur in a significant percentage of these patients. Recurrent implantation failure (RIF) can be defined as the absence of a positive pregnancy after multiple/repeated good-quality embryo transfers in women under the age of 40. Implantation failure is diagnosed by embryo failure to progress to a stage where intrauterine gestational sac is ultrasonography identified [3,4]. However, there is no consensus on the definition for such condition [5]. RIF is associated with a variety of factors, such as maternal age, lifestyle, obesity, genetic disorders or immunological conditions [6].

Endometrial receptivity is considered one of the most critical prognostic markers in the success of a pregnancy following embryo transfer [7]. Endometrial thickness of <7 mm has been linked to poor pregnancy outcomes, RIF or high rate of cycle cancelation rate [8,9,10,11,12]. Recent studies suggest that the embryonic implantation process is highly dependent on interleukins, growth factors or cytokines among others [13,14,15]. Recently, different therapies have been assessed to treat infertility such as estradiol hormonal supplementation [16], low-dose aspirin [17,18], vitamin E [19], vaginal sildenafil [20,21,22], pentoxifylline [19], tamoxifen [23] and stem cell therapy [24]. In spite of these therapeutic alternatives, many patients still are suffering from implantation failure, especially women with a thin endometrium (TE). Therefore, there is a need for new therapeutic approaches to the treatment of women with poor responsiveness to conventional ART [25,26].

Autologous platelet concentrates have emerged as an alternative to promoting endometrium receptivity and enhancing pregnancy outcomes. Of the available formulations, platelet-rich plasma (PRP) has been applied in multiple medical fields including oral and maxillofacial surgery [27], dermatology [28], traumatology [29], ophthalmology [30], and, most recently, reproductive medicine. PRP is prepared by gradient density centrifugation of blood to obtain a concentrated platelets in plasma. Following platelet activation, platelets granules release their contents.

In the last years, several controlled trials have been published, evaluating the efficacy of intrauterine PRP infusion over nonintervention therapy. However, whether PRP is an effective treatment in women with RIF or TE remains controversial. Some studies described effects of PRP that were generally positive in terms of endometrial thickness and pregnancy outcomes, but no meta-analysis that specifically evaluates randomized clinical trials (RCT) has been conducted so far [11,31,32]. According to in vitro data, the beneficial effects of PRP on endometrial cells have enhanced their response (proliferation, migration and differentiation) as well as an angiogenesis [33,34]. Interestingly, favorable outcomes have also been reported in patients with low ovarian reserve [35], premature ovarian insufficiency [36] and post-menopausal women [36] after intraovarian PRP administration. By contrast, some researchers observed no benefit for embryo implantation, pregnancy rate or live birth rate when PRP was administered in the uterus [37,38]. The aim of our study was to investigate whether intrauterine PRP infusion can improve embryo implantation and its potential for enhancing pregnancy outcomes in women with a history of ART failure, through an analysis of the available RCTs.

## 2. Materials and Methods

### 2.1. Protocol Registration and Reporting Format

This systematic review with meta-analysis was conducted according to the Preferred Reporting Items for Systematic Reviews and Meta-Analyses (PRISMA) guidelines [39]. The protocol was registered and assigned in the PROSPERO database (CRD42021287963).

### 2.2. Focus Question

This review aimed to answer the following question:

Does intrauterine infusion of PRP improve pregnancy outcomes in women with a history of embryo transfer failure?

### 2.3. PICO Strategy

The following strategy was constructed according to PICO study design:The participants (P) included were women undergoing assisted reproduction with a history of embryo transfer failure;The intervention (I) was intrauterine PRP infusion before embryo implantation;The comparison (C) was to no intervention or placebo;The outcomes (O) were implantation rate, biochemical pregnancy rate, clinical pregnancy rate, live birth rate and miscarriage rate.

### 2.4. Eligibility Criteria

Participants included in the study had to be healthy women with no underlying severe diseases. Subjects had a history of failed embryo transfer and there was not a minimum number of participants or a limit on the number of patients treated.

The inclusion criteria had been: (a) study on humans only, (b) intervention of PRP by intrauterine administration before embryo implantation, (c) comparator was no intervention or placebo, and (d) study was designed as RCT.

The following publications were excluded: (a) observational studies; (b) trials with inadequate information or insufficient information regarding selected topic.

### 2.5. Data Sources and Search Strategy

The following databases were used for the systematic search from inception to 19 August 2022: MEDLINE/PubMed, Cochrane Central Register of Controlled Trials and Ovid. The following keywords were used in the search strategy: (Platelet-rich plasma OR platelet-rich fibrin OR plasma rich in growth factors OR PRP OR PRF OR PRGF) AND (infertility OR pregnancy OR endometrium OR thin endometrium OR endometrial thickness OR embryo transfer OR ovary). The search did not apply any language restriction. A screening of other systematic reviews was also carried out for possible additional trials.

### 2.6. Study Selection

Two reviewers independently (M.A. and M.d.l.F.) screened the search hits (the title and abstract) to exclude studies that had not met the research question and selection criteria. In case of disagreement, discussion was implemented to reach consensus. Then, full texts of the eligible articles or those with insufficient information in the title and abstract were obtained. The reviewers independently selected the studies that met the selection criteria. A third reviewer (M.H.A.) solved disagreement between the reviewers. The excluded publications were listed with the reason for exclusion (Appendix A).

### 2.7. Data Extraction

Two authors (M.A. and M.d.l.F.) independently extracted relevant data on study, participants and PRP protocol and characteristics into a specific Excel (Microsoft) spreadsheet. Discrepancies were eliminated by referring back to the full-text of the article. The extracted data were: (a) study characteristics—primary author, year of publication, number of participants and country; (b) patient characteristics—age and cause of failed embryo transfer; (c) PRP characteristics—obtention protocol, anticoagulant, presence of leukocytes, platelet concentration, PRP activation, method of application, dose, time to embryo transfer; (d) outcome assessment—implantation rate, biochemical pregnancy rate, clinical pregnancy rate, live birth rate and miscarriage rate. One of the reviewers (M.A.) included the data in Review Manager 5.4. Data accuracy was double-checked.

### 2.8. Risk of Bias Assessment

The methodologic quality of each trial was investigated by using risk of bias assessment in accordance with the Cochrane Handbook for Systematic Reviews of Interventions [40]. Each trial was classified at high, low or unclear risk of bias. Two authors independently evaluated the selected studies.

Using the Cochrane Risk of Bias Tool, each study was classified at high, moderate or low risk of bias. If none of the six domains were found to be at high risk, and if three or fewer domains were found to be at unclear risk, an overall low risk rating was assigned. An overall rating of moderate risk was attributed when one of the domains was high risk; or when no domain was high risk, but four or more were unclear risk. In all other cases, the study was considered to have an overall high risk of bias.

### 2.9. Outcomes

The primary outcome was clinical pregnancy rate, whereas secondary outcomes were implantation rate, biochemical pregnancy rate, live birth rate and miscarriage rate. All outcomes were defined according to the study authors (Appendix A).

### 2.10. Quality of Evidence

Two authors (M.A. and M.H.A.) independently evaluated the overall certainty of evidence using the grades recommendation, assessment, development and evaluation (GRADE) approach [40]. GRADEpro Guideline Development Tool (McMaster University, Hamilton, Canada, 2020, developed by Evidence Prime, Inc., Krakow, Poland, available from https://www.gradepro.org/, accessed on 8 February 2023) was used to assess the quality of the body of retrieved evidence [41].

### 2.11. Statistical Analysis

The software Review Manager 5.4 (The Nordic Cochrane Centre, Copenhagen, Denmark) was used to conduct the meta-analysis. Outcomes were assessed as dichotomous variables using the Mantel–Haenszel method and recorded as risk ratio (RR) with 95% confidence interval (CI). A pooled RR was interpreted as follows: a value higher than 1 means that the event occurs more in the exposure group. However, a RR value inferior to 1 indicates that the event occurs less in the exposure group.

Heterogeneity among the selected studies was evaluated using the I2 statistic (substantial heterogeneity is indicated by a value > 50%). This allowed to select either a fixed effects model (heterogeneity was not significant) or a random effects model (heterogeneity was significant). When significant heterogeneity was identified, the random effects model was used. The results were shown in a forest plot of interventions. Sensitivity analyses of reporting bias by funnel plots were not performed (the small number of studies).

## 3. Trial Sequential Analysis

Trial Sequential Analysis (TSA) was undertaken for the primary and secondary outcomes, to estimate the power of the meta-analysis results and to consider types I and II errors. The software TSA 0.9.5.10 Beta (Copenhagen Trial Unit Centre for Clinical Intervention Research Department, Copenhagen, Denmark) was used. Fixed or random effects model was chosen for meta-analysis, as appropriate. The 95% confidence intervals for inconsistency (I^2^) were also estimated with TSA software. The type I and type II errors were set at 5% and 20% (80% power), respectively, to estimate the required information size (RIS) and alpha monitoring limits. For the calculation of the RIS, the incidence rates in the test (PRP) and control arms were estimated in accordance with the results of the meta-analysis. No correction for heterogeneity was applied. Graphical analysis showed whether the cumulative Z curve (blue) crossed the trial sequence monitoring threshold (horizontal red line) and RIS threshold (vertical red line).

## 4. Results

### 4.1. Summary of the Literature Search

The Flow chart of the selected studies is depicted in Figure 1. The literature search yielded 2053 publications. The titles and abstracts were then screened, resulting in 26 studies possibly eligible selection. After reading the full-text publications, 14 studies were excluded (Appendix A). Qualitative synthesis was performed with the remaining 12 studies. Two studies did not provide enough data and were excluded from the quantitative me-ta-analysis [42,43].

### 4.2. Study Characteristics

In this systematic review, only RCTs that compared intrauterine PRP infusion before embryo transfer to no intervention or placebo group were included. Participants were women with a history of RIF [31,32,37,42,43,44,45,46,47] or TE [11,12]. Table 1 describes the principal characteristics of the selected studies. All trials were conducted between 2017 and 2022, of which eight studies were published after 2020. Eleven out of twelve studies were conducted in Iran and one in Russia. The number of patients in each study varied from 40 to 393. Hormone replacement therapy (HRT) was applied to all participants, regardless the experimental group. Nine studies infused ≤1 mL PRP [11,12,31,32,37,42,44,45,46], and two studies administered >1 mL [38,47], whereas one trial did not provide this information [43]. None of the studies reported on the activation of the PRP before application. The following outcomes were evaluated: implantation rate, biochemical pregnancy rate, clinical pregnancy rate, live birth rate and miscarriage rate (Table 2).

### 4.3. Risk of Bias of Included Trials

From a total of twelve studies, ten were selected for the quantitative analysis and subjected to a risk of bias assessment (Figure 2). The item of selection bias was judged low in eight studies that provided a random sequence generation description, while it was unclear in the remaining two studies. The item of selection bias in relation to allocation concealment was judged low risk of bias in seven studies and unclear risk of bias in three studies. The item performance bias was judged low risk of bias in three studies and unclear risk of bias in seven studies. The attrition bias was judged low risk of bias in nine studies and unclear risk of bias in one study (it was unclear in one study whether the follow-up reports were completed). The reporting bias was judged low risk of bias in three studies, unclear risk of bias in five studies and high risk of bias in two studies (did not report all planned outcomes). 

After the assessment, the overall risk of bias was low in seven studies, moderate in one study and high in two studies.

**Table 2 bioengineering-10-00303-t002:** Outcomes of the selected studies.

Study	Sample Size		Implantation Rate		Biochemical Pregnancy Rate		Clinical Pregnancy Rate		Live Birth Rate		Miscarriage Rate	
	Control	PRP	Control	PRP	Control	PRP	Control	PRP	Control	PRP	Control	PRP
Allahveisi et al., 2020	25	25	36% (*n* = 9)	28% (*n* = 7)	NR	NR	36% (*n* = 9)	28% (*n* = 7)	28% (*n* = 6)	24% (*n* = 7)	NR	NR
Eftekhar et al., 2018	43	40	9.37% *	21% *	24.2% (*n* = 8)	42.4% (*n* = 14)	18.2% (*n* = 6)	39.4% (*n* = 13)	18.2% (*n* = 6)	33.3% (*n* = 11)	6% (*n* = 2)	9% (*n* = 3)
Ershadi et al., 2022	45	40	0.38 ± 0.16%	0.36 ± 0.24%	27% (*n* = 12)	40% (*n* = 16)	24% (*n* = 11)	33% (*n* = 13)	NR	NR	31.25% (*n* = 5)	8.33% (*n* = 1)
Nazari et al., 2020	48	49	NR	NR	27.08% (*n* = 13)	53.06% (*n* = 26)	16.66% (*n* = 8)	44.89% (*n* = 22)	NR	NR	NR	NR
Nazari et al., 2019	30	30	NR	NR	6.7% (*n* = 2)	40% (*n* = 12)	3.3% (*n* = 1)	33.3% (*n* = 10)	NR	NR	NR	NR
Nazari et al., 2021	197	196	NR	NR	24.87% (*n* = 49)	51.53% (*n* = 101)	19.28% (*n* = 38)	48.97% (*n* = 96)	5.58% (*n* = 11)	39.28% (*n* = 77)	NR	NR
Nazari et al., 2022	20	20	NR	NR	NR	NR	20% (*n* = 4)	35% (*n* = 7)	0% (*n* = 0)	15% (*n* = 3)	20% (*n* = 4)	20% (*n* = 4)
Obidniak et al., 2017	45	45	20.9% *	40.5% *	NR	NR	24.4% (*n* = 11)	53.3% (*n* = 24)	NR	NR	NR	NR
Zamaniyan et al., 2020	43	55	34.9% (*n* = 15)	63.6% (*n* = 35)	23.3% (*n* = 10)	36.2% (*n* = 20)	23.3% (*n* = 10)	52.7% (*n* = 29)	NR	NR	4.6% (*n* = 2)	1.8% (*n* = 1)
Zargar et al., 2021	40	40	5% (*n* = 2)	15% (*n* = 6)	NR	NR	2.5% (*n* = 1)	12.5% (*n* = 5)	2.5% (*n* = 1)	12.5% (*n* = 5)	2.5% (*n* = 1)	2.5% (*n* = 1)

* Data excluded from the meta-analysis; NR: not reported.

### 4.4. Quality of Evidence

GRADE was used to qualify the evidence of the study outcomes (Table 3). Regarding to the studies with RIF patients, we downgraded the certainty of the implantation rate by two levels to low due to very serious limitation in the imprecision (optimal information size is not met and the 95% CI of the relative risk (RR) included RR of 1.25). The same was for the certainty of live birth rate and miscarriage rate. Remaining outcomes (biochemical pregnancy rate and clinical pregnancy rate) were judged as high certainty of evidence. With regard to TE studies, the level of evidence of biochemical pregnancy rate was downgraded to low due to the unclear risk of bias in one study, the optimal information size was not met, and the RR included a value of 1.25 (Table 4). The same limitations applied to the clinical pregnancy outcome The certainty of the outcomes of the live birth rate and miscarriage rate was downgraded by three levels to very low due to the unclear risk of bias in one study, the optimal information size was not met, the 95% CI of the RR included a RR of 0.75 or 1.25 and a wide range of the 95% CI of the RR (miscarriage rate). 

### 4.5. Effect of PRP on Clinical Pregnancy Rate

Ten studies evaluated clinical pregnancy in 933 women with RIF [31,32,37,38,44,45,46,47] and 143 women with TE [11,12]. Clinical pregnancy was interpreted as the presence of fetal heartbeat in transvaginal ultrasound 5–6 weeks after embryo transfer in six studies [11,12,31,32,45,46]. Other study measured clinical pregnancy rate by dividing the number of fetal poles with an observed heartbeat in the 6-week-old sonogram by the number of the transferred embryos [37]. Other study defined it as the presence of an embryonic sac at 5–6 weeks gestation [44]. Two publications did not define the term [38,47].

A total of 325 pregnancies were reported; the percentage of pregnancies was 41.85% (226 out of 540) and 18.47% (99 out of 536) in the PRP and control groups, respectively (Appendix A). 

The meta-analysis including eight studies (excluding high-risk bias studies) showed higher pregnancy rate when intrauterine PRP was administered (RR: 2.19 (95% CI: 1.78 to 2.70), I^2^: 30%). In this line, statistical trend was maintained when RIF studies were analyzed separately (RR: 2.18 (95% CI: 1.76 to 2.70), I^2^: 40%). No meta-analysis could be performed for the TE sub-group (Figure 3A). 

### 4.6. Effect of PRP on Implantation Rate 

Five studies analyzed implantation rate after PRP treatment [12,37,38,46,47]. These studies were composed of a total of 401 infertile women who had experienced RIF (four studies) or TE (one study). Implantation rate was defined as the ratio of gestational sacs to the number of embryos transferred in three trials [12,37,46]; authors did not provide a detailed description in two publications [38,47].

Out of four trials investigating women with RIF, two studies reported a beneficial effect for PRP [46,47], whereas no statistical differences were found in other two trials [37,38]. In this line, significative difference between groups was observed in the trial composed of a TE cohort [12]. Due to missing data, two studies were finally excluded from the meta-analysis [12,47]. The remaining studies corresponded to women with RIF and showed a better performance in terms of implantation rate when PRP was administered before embryo transfer RR: 1.57 (95% CI: 1.07 to 2.30), I2: 0% (Figure 3B). No meta-analysis could be performed for the TE sub-group.

### 4.7. Effect of PRP on Biochemical Pregnancy Rate

Six studies including 816 participants (410 cases and 406 controls) evaluated the effect of intrauterine PRP infusion before embryo transfer on biochemical pregnancy rate [11,12,31,32,45,46]. Four studies included infertile women who had experienced RIF [31,32,45,46], whereas TE pathology was studied in two trials [11,12]. All studies defined biochemical pregnancy as a positive serum β-human chorionic gonadotropin after 14 days from embryo transfer.

Women subjected to PRP treatment showed a biochemical pregnancy rate of 46.10% (189 out of 410); on the contrary, the control group displayed a 23.15% (94 out of 406) rate (Appendix A). The meta-analysis, including five studies (excluding high-risk bias study), showed a significant positive effect for the PRP group in comparison to conventional treatment (RR: 1.91 (95% CI: 1.55 to 2.35), I^2^: 0%). When studies were evaluated separately by type of pathology, a beneficial effect was only observed for the PRP in women with RIF (RR: 1.91 (95% CI: 1.54 to 2.37), I^2^: 0%) (Figure 3C).

### 4.8. Effect of PRP on Live Birth Rate

Five RCTs composed of a total of 646 patients assessed the effect of PRP on live birth rate [12,32,37,38,44]. Four trials were conducted on women presenting a history of RIF [32,37,38] and one study was performed on women with TE [12]. Four studies considered birth after 20 to 24 weeks of gestation as a live birth [12,37,38,44]. Authors did not provide a clear definition in another trial [32].

Live birth rate reached to 32.09% (103 out of 321) when PRP treatment was applied, whereas the rate was reduced to 7.38% (24 out of 325) in its absence. Four out of five studies reported a nonsignificant difference [12,37,38,44] (Appendix A). However, the overall meta-analysis (excluding one high-bias study) showed better outcomes following a PRP-based protocol (RR: 2.90 (95% CI: 1.06 to 7.96), I^2^: 77%). Analyzing RIF studies separately, the risk ratio increased to 3.36 (95% CI: 0.84 to 13.45), I^2^: 81% (Figure 3D).

### 4.9. Effect of PRP on Miscarriage Rate

Five studies evaluated the occurrence of miscarriages following intrauterine PRP infusion [12,38,44,45,46]. From a total of 386 participants, PRP was administered to 195 women (Appendix A). Two studies [12,44] considered a miscarriage as pregnancy loss before 20 weeks of gestation. A clear definition was absent in the other three trials [38,45,46].

Our meta-analysis (excluding one high-risk bias study) did not reveal a significant effect of PRP on the miscarriage rate (RR: 1.68 (95% CI: 0.64 to 4.42), I^2^: 0%). Nonsignificant differences remained when a subgroup analysis was performed in RIF patients (RR: 1.71 (95% CI: 0.54 to 5.48), I^2^: 30%). Only one trial evaluated miscarriages on TE cohort and no meta-analysis could be performed (Figure 3E).

### 4.10. Trial Sequential Analysis

Trial sequential analysis was only undertaken for RIF, as many subgroups with TE had insufficient data to perform the analysis. TSA showed that the cumulative Z-curve was maintained well beyond the significance threshold in favor of the PRP group for all outcomes, except for miscarriage rate. In addition, the total sample size for RIF was above the required information size (RIS) for biochemical pregnancy rate (RIS = 199), clinical pregnancy rate (RIS = 234) and live birth rate (RIS = 331), but it was inferior for implantation rate (RIS = 806) and for miscarriage rate (RIS = 3756). This suggested that the meta-analysis had sufficient power to detect the beneficial effect of PRP for biochemical pregnancy rate, clinical pregnancy rate and live birth rate. Conversely, more studies are required to achieve significance regarding the advantageous effect of PRP for implantation rate and, especially, for miscarriage rate. From the available evidence regarding the last outcome, there is the possibility that additional studies might confirm the conclusion that PRP does not provide advantages for miscarriage rate.

## 5. Discussion

During the last decades, a variety of therapeutic approaches and interventions have been proposed to manage poor responsiveness to in vitro fertilization techniques, including uterine interventions [48,49,50,51,52], gynecological surgical procedures [53,54,55,56,57,58,59,60], immunomodulatory therapies [61,62,63,64,65,66,67,68] or treatments to enhance endometrial receptivity [69,70,71,72,73]. However, it was not until 2015 [8] that autologous platelet concentrates were used to promote endometrial growth and improve pregnancy outcome in infertile women with TE. Since then, the available clinical studies are increasing rapidly and PRPs are becoming the treatment of choice for many physicians. The present systematic review investigated the available literature to analyze the ability of PRP to manage infertility problems in women with a history of embryo transfer failure.

Ten RCTs were selected for our meta-analysis and a total of 540 women receiving intrauterine PRP infusion were included. According to TSA results, our meta-analysis fulfilled the information size requirement providing sufficient evidence for biochemical pregnancy rate, clinical pregnancy rate and live birth rate outcomes. To our knowledge, there are currently no published meta-analyses in the field of reproduction that specifically conduct a TSA to estimate the level of evidence. Conversely, information size requirement was found to be insufficient for implantation rate and miscarriage rate. This finding can be explained by the small number of patients recruited for this outcome and the high heterogeneity observed among studies. Future clinical trials and high-standardized protocols for PRP preparation and infusion are needed.

Since its discovery, PRP has been prescribed in the field of regenerative medicine for a wide variety of pathologies, including dermatology [74], maxillofacial surgery [75,76], traumatology [77,78] or in vitro reproduction [79,80,81] among others. Only two systematic reviews with meta-analysis have been published evaluating the beneficial effect of PRP in assisted reproduction. The first study was published by Maleki-Hajiagha et al. in 2020 and reported favorable outcomes for PRP group in women undergoing assisted reproduction when comparing to nonintervention control subjects [82]. More recently, Busnelli and colleagues performed a meta-analysis analyzing all available treatments and interventions in women with RIF [83]. In agreement with the first review, a beneficial effect was observed in terms of successful pregnancies when autologous PRP was infused before embryo transfer. The overall conclusion reached by these studies was similar to the present review, showing that PRP therapy confers additional benefits over conventional hormone replacement treatments. Nevertheless, the number of RCTs included in our systematic reviews is markedly higher compared to the previous ones. Moreover, no strength of evidence assessment was attempted, and no TSA analysis was presented to evaluate the power of the meta-analysis. In addition, nonrandomized controlled trials were also included in previously published systematic reviews. These types of studies might be an important source of bias.

Autologous PRP can be manufactured using different protocols that might lead to products with different composition and functionality [76]. It is likely that PRP composition plays a pivotal role in the results of the trial. In fact, the presence or absence of leukocyte is considered a key differentiator for many authors [76,84]. Unfortunately, none of the selected studies provided an accurate description of leukocyte concentration, and two trials did not even mention whether leukocytes were present or not. The different methodology used to prepare PRP (manufacturer, a comprehensive protocol description or leukocyte and platelet counts, among others) should be clearly provide by the authors, as these factors might have a clear impact on the measured clinical outcomes.

According to the published studies, the prevalence of TE ranged from 2.4% to 8.5% [85,86,87]. The management of women with insufficient endometrial thickness or a refractory endometrium is one of biggest challenges in assisted reproduction. In the last years, PRP therapies have emerged as a cutting-edge treatment to promote endometrial growth and enhance the success of embryo transfer. However, meta-analysis for TE studies could not be performed due to the exclusion of high-risk bias studies (leaving only one study). The current literature establishes 7 mm as the minimum endometrium thickness needed to accomplish the embryo transfer. Three of the selected studies administered a second intrauterine PRP infusion if the thickness did not reach 7 mm [11,12,38]. However, a meta-analysis suggests the cutoff of 7 mm has limited prognostic value for clinical pregnancy [88] and questions the requirement of this second infusion. Future studies should clarify the clinical usefulness of this thickness limit.

Another critical parameter for autologous PRP performance is the dose infused in the uterus. In the meta-analysis performed by Maleki-Hajiagha et al. [82], a dose of 0.5–1 mL showed a higher effect in comparison to doses of <0.5 mL in terms of clinical pregnancy. Our study did not consider this factor in the meta-analysis; however, a qualitative assessment of the data showed a high performance for both doses. The standardization of the PRP dose should be the focus of future clinical trials. 

Although autologous PRP are being adopted by many fertility clinics, the data describing the safety profile are still limited. Only five out of the ten selected studies reported the incidence of miscarriage after PRP infusion. More importantly, none of the trials monitored PRP-related adverse events in the months following the childbirth or their effect on conceived children. Consequently, this aspect could be an interesting topic for upcoming clinical trials. PRP has been used in the context of in vitro fertilization to treat patients with repeated pregnancy loss. However, the level of the available evidence is not sufficient to propose the use of in vitro fertilization to treat patients with unexplained recurrent pregnancy loss [89].

The number of clinical trials using PRP in the field of reproduction is increasing rapidly. However, the underlying molecular mechanism of PRP to enhance endometrial receptivity has not been clearly described. In view of the experience derived from other medical fields, it is likely that PRP plays a pivotal role in endometrium tissue growth and regeneration by releasing growth factors and cytokines. Consequently, a higher migration and proliferation rate have been observed on different human endometrial cells following PRP application [7]. In this line, increased levels of the proliferative marker Ki67 have been measured in the endometrial stromal cell line ICE7 [90] and uterine horn [91]. More recently, the expression of Ki67 and homeobox A10 (Hoxa10) was found to be upregulated in the endometrium of mice subjected to a uterine damage model [92]. The potential role of PRP in inflammation has also been extensively studied. In fact, anti-inflammatory properties of PRP involve the downregulation of several key players in the inflammatory pathway [93], including the nuclear factor kappa-B (NF-κB) [94]. In addition, PRP is able to modulate a wide array of chemokines such as chemokine ligand 5 (CCL5) [7], lipoxin A4 [94], IL-1B, IL-8, TNF-A, COX-2 or iNOS [95]. Some authors have speculated these factors result in a reduced intrauterine fluid accumulation and, therefore, diminishing inflammation [94,96]. Other evidence suggests decreased fibrosis and a restoration of endometrial structure following PRP administration in a murine model of uterine injury [97]. A reduced expression in collagen type 1A, transforming growth factor β1 (TGF-β1) and tissue inhibitor of metalloproteinase 1 (TIMP1) were reported, supporting the notion that PRP might promote endometrial regeneration when the endometrium is damaged. Although there is a lack of studies associating directly immune response to PRP infusion in the endometrium, it is well known that platelets participate in innate and adaptive immunity [98,99], with the ability to interact with many immune cells. The interaction of platelets with neutrophils occurs via P-selectin/PSGL-1 and, subsequently, is able to induce leukocyte signaling and neutrophil integrins activation to promote degranulation [100,101]. The initial contact between platelets and monocyte also involves P-selectin and PSGL-1, and this interaction enhance expression of β1-integrin, β2-integrin and BF-κB pathways [102]. Overall, evidence of the underlying mechanism of PRP is still sparce, and it is likely that several factors and molecular pathways are involved (Figure 4). 

The strengths of this systematic review included a restriction to RCTs to avoid biases from observational studies, the assessment of bias with the Cochrane Handbook for Systematic Reviews of Interventions [40], the evaluation of the strength of evidence with the grade approach [41] and the implementation of TSA to address information size requirement for each outcome [103]. In addition, most of the studies used similar outcomes definitions and the follow-up time was comparable among studies, avoiding biases caused by the different methodology or trial design. 

However, several limitations need to be considered. First, the search strategy did not incorporate MeSH (Medical Subject Headings) terms. Second, there is no standardized protocol for PRP products, with some preparations including leukocytes among their composition and, therefore, conclusions should be interpreted with caution. Third, most of available trials were performed in the same country (Iran), the ethnicity being an important confounding factor that can lead to a data misinterpretation. Fourth, some publications reported an additional infusion of PRP when endometrium thickness was inadequate for embryo implantation. This factor was not considered in the meta-analysis due to the absence of information provided by the authors.

The present systematic review demonstrates that PRP might be an effective therapeutic approach in cases of poor responsiveness to conventional treatments in assisted reproductive techniques. Based on our meta-analysis and the subsequent TSA evaluation, autologous PRP could enhance biochemical pregnancy rate, clinical pregnancy rate and live birth rate. Additional well-designed trials are needed to confirm the beneficial effect of PRP over the implantation rate and miscarriage rate. Accordingly, there is lack of evidence to draw conclusions on whether PRP can bring benefit on women with TE due to the limited number and quality of the studies. Future studies should also elucidate the safety profile of intrauterine PRP administration with long-term follow-ups.

## Figures and Tables

**Figure 1 bioengineering-10-00303-f001:**
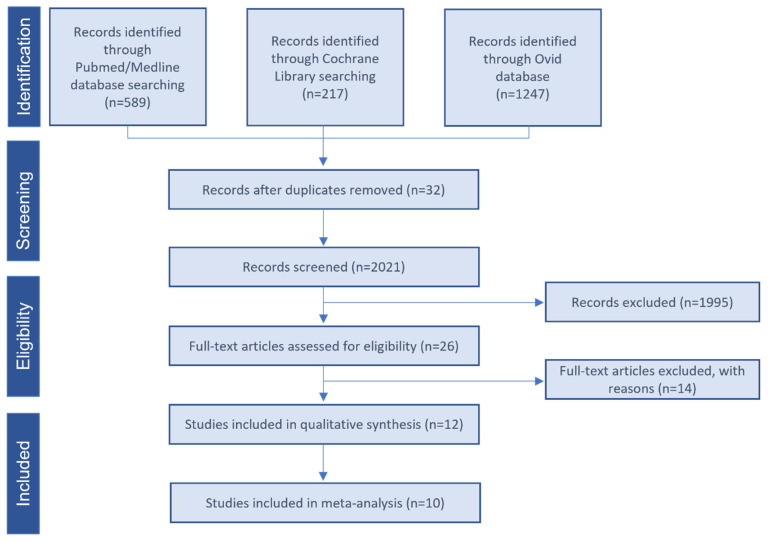
Study selection flow diagram. PRISMA flow diagram of the screening and selection process.

**Figure 2 bioengineering-10-00303-f002:**
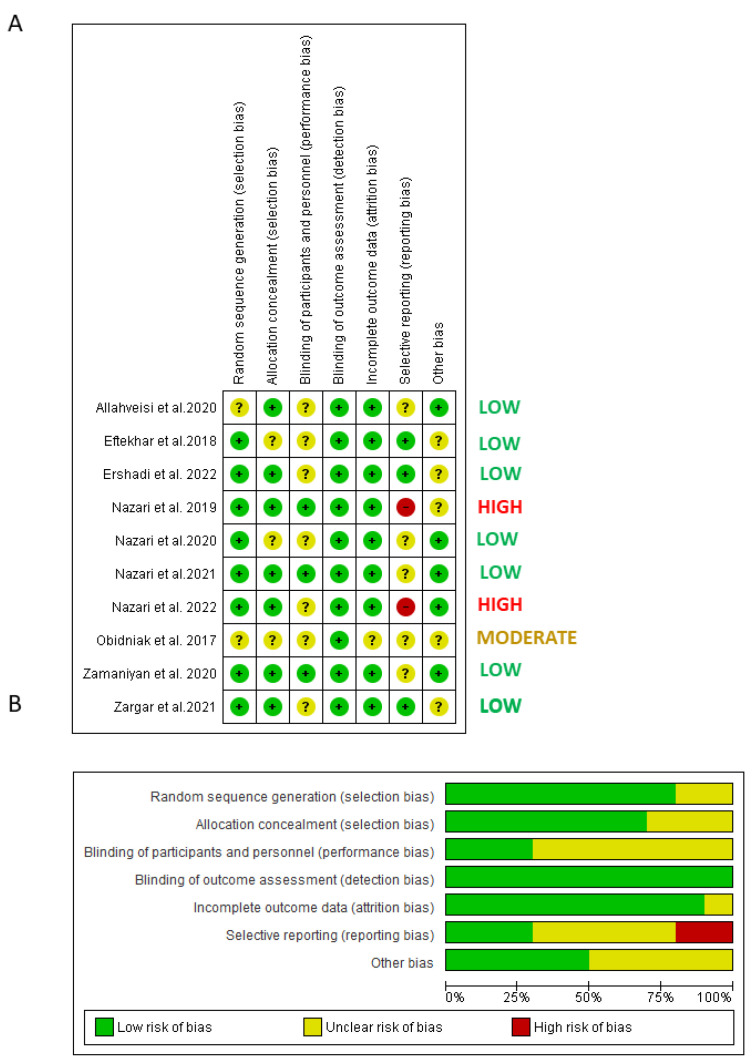
Quality assessment of the included RCTs. (**A**) Risk of bias summary: review authors’ judgments about each risk of bias item for each included study: (+), low risk of bias; (−): high risk of bias; (?): unclear risk of bias. (**B**) Risk of bias graph: review authors’ judgments about each risk of bias item presented as percentages across all included studies.

**Figure 3 bioengineering-10-00303-f003:**
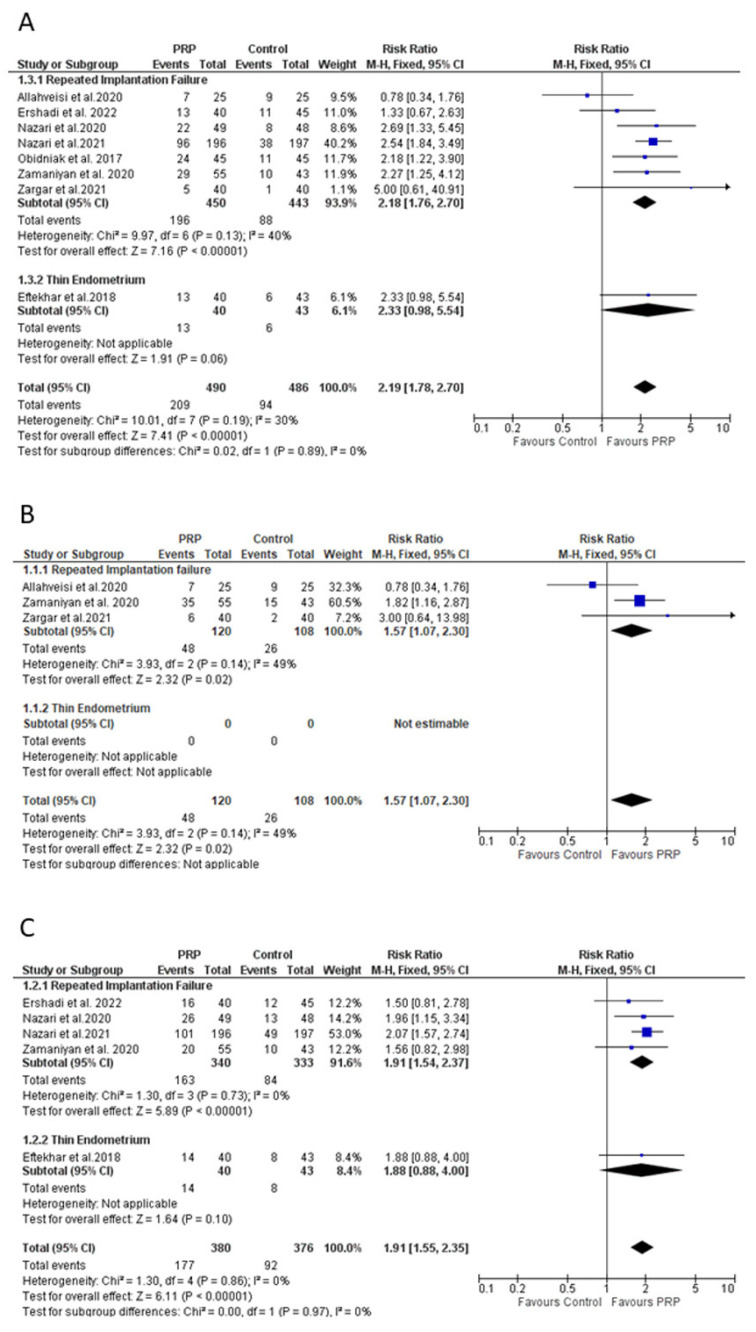
Meta-analysis of the studies evaluating (**A**) clinical pregnancy rate, (**B**) implantation rate, (**C**) biochemical pregnancy rate, (**D**) live-birth rate, and (**E**) miscarriage rate. CI: confidence interval; PRP: platelet-rich plasma. Studies with high-risk bias were excluded.

**Figure 4 bioengineering-10-00303-f004:**
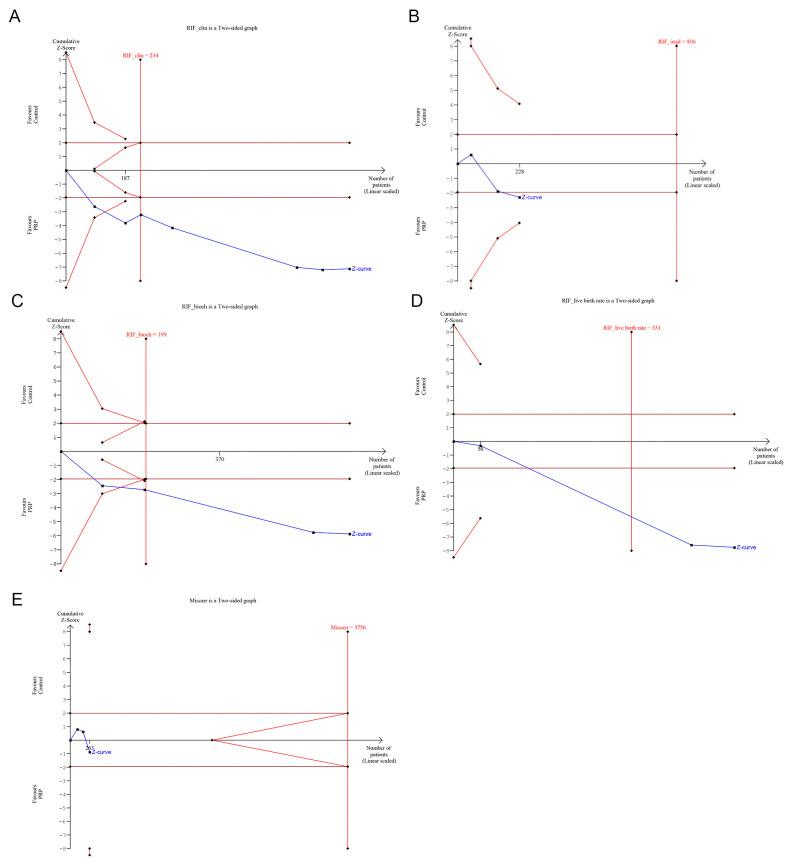
Trial sequential analysis for (**A**) clinical pregnancy rate, (**B**) implantation pregnancy rate, (**C**) biochemical pregnancy rate, (**D**) live birth rate and (**E**) miscarriage rate in recurrent implantation failure (RIF) studies. Studies with high-risk bias were excluded.

**Table 1 bioengineering-10-00303-t001:** Main characteristics of the selected studies.

Study	Sample Size		Age (Years)		Cause of Failed Embryo Transfer	PRP Obtention Protocol	Anticoagulant	Leukocytes	Platelet Concentration	Intervention (Volume of PRP)	PRP Activation	Application Method	Embryo Transfer (Time after PRP)	Second PRP Instillation
	Control	PRP	Control	PRP										
Allahveisi et al., 2020	25	25	33.8 ± 0.54	33 ± 0.9	RIF	1700 g 12 min; 3300 g 7 min	ACD-A	Y	411 × 10^3^–1067 × 10^3^/uL	0.5 mL	NR	Intrauterine infusion	48 h	No
Bakhsh et al., 2021	50	50	32.7	35	RIF (≥4)	1400 rpm 10 min; 3500 rpm 6 min	ACD	N	4–5 times	0.5 mL	NR	Intrauterine infusion under ultrasound guidance	48 h	NR
Eftekhar et al., 2018	43	40	32.4 ± 2.63	31.98 ± 2.26	Poor endometrial response (ET < 7 mm) to HRT	1600 g 10 min; 3500 g 5 min	ACD-A	Y	4–5 times, 2000 lymphocyte	0.5–1 mL	NR	Intrauterine infusion	When ET ≥ 7 mm	Yes (if ET < 7 mm)
Ershadi et al., 2022	45	40	31.2 ± 4.8	31.3 ± 4.3	RIF	1200 rpm 12 min; 3300 rpm 7 min	Citrate acid	Y	4–5 times	0.5 mL	NR	Intrauterine infusion under ultrasound guidance	48 h	NR
Ghasemi et al., 2020	85		NR	NR	RIF	NR	NR	NR	NR	NR	NR	Intrauterine infusion	48 h	NR
Nazari et al., 2020	48	49	34.95 ± 4.23	35.73 ± 3.49	RIF (≥3)	1200 rpm 10 min; 3300 rpm 5 min	ACD-A	Y	4–5 times	0.5 mL	Not clear	Intrauterine infusion under ultrasound guidance	48 h	No
Nazari et al., 2019	30	30	32.33 ± 4.79	33.93 ± 2.76	Thin endometrium (ET < 7 mm)	1200 rpm 12 min; 3300 rpm 7 min	ACD-A	Y	NR	0.5 mL	NR	Intrauterine infusion under ultrasound guidance	48 h	Yes (if ET < 7 mm)
Nazari et al. 2021	197	196	33.61 ± 4.06	34.11 ± 3.75	RIF (≥3)	1200 rpm 12 min; 3300 rpm 7 min	NR	Y	4–5 times	0.5 mL	NR	Intrauterine infusion	48 h	No
Nazari et al. 2022	20	20	34.75 ± 4.57	35. 70 ± 5.10	RPL	1200 rpm 12 min; 3300 rpm 7 min	ACD-A	Y	4–5 times	0.5 mL	NR	Intrauterine infusion under ultrasound guidance	48 h	NR
Obidniak et al., 2017	45	45	28–39		RIF	Plasmolifting technology	NR	NR	NR	2 mL	NR	Intrauterine infusion	NR	NR
Zamaniyan et al., 2020	43	55	33.13 ± 5.00	33.88 ± 6.32	RIF (≥3)	1200 rpm 12 min; 3300 rpm 7 min	ACD-A	Y	4–7 times	0.5 mL	NR	Intrauterine infusion	48 h	No
Zargar et al., 2021	40	40	32.82 ± 5.18	34.15 ± 5.14	RIF (≥2)	12,000 g 10 min; 12,000 g 10 min	ACD	N	NR	1.5 mL	NR	Intrauterine infusion	48 h	Yes (if ET < 7 mm)

RIF: repeated implantation failure; HRT: hormone replacement therapy; ET: endometrial thickness; NR: not reported; ACD: anticoagulant citrate dextrose solution; RPL: recurrent pregnancy loss.

**Table 3 bioengineering-10-00303-t003:** Summary of the quality assessment by GRADE approach of outcomes included in the meta-analysis of repeated implantation failure patients.

Certainty Assessment							No. of Patients		Effect		Certainty	Importance
No. of Studies	Study Design	Risk of Bias	Inconsistency	Indirectness	Imprecision	Other Considerations	PRP	Conventional Treatment	Relative (95% CI)	Absolute (95% CI)		
**Implantation rate**												
3	Randomized trials	Not serious	Not serious	Not serious	Very serious ^a^	None	48/120 (40.0%)	26/108 (24.1%)	**RR 1.57** (1.07 to 2.30)	**137 more per 1000** (from 17 more to 313 more)	⨁⨁◯◯ Low	IMPORTANT
**Biochemical pregnancy rate**												
4	Randomized trials	Not serious	Not serious	Not serious	Not serious	None	163/340 (47.9%)	84/333 (25.2%)	**RR 1.91** (1.54 to 2.37)	**230 more per 1000** (from 136 more to 346 more)	⨁⨁⨁⨁ High	CRITICAL
**Clinical pregnancy rate**												
7	Randomized trials	Not serious	Not serious	Not serious	Not serious	Strong association	196/450 (43.6%)	88/443 (19.9%)	**RR 2.18** (1.76 to 2.70)	**234 more per 1000** (from 51 more to 338 more)	⨁⨁⨁⨁ High	CRITICAL
**Live birth rate**												
3	Randomized trials	Not serious	Serious ^b^	Not serious	Very serious ^c^	Strong association	89/261 (34.1%)	18/262 (6.9%)	**RR 3.36** (0.84 to 13.45)	**162 more per 1000** (from 11 more to 855 more)	⨁⨁◯◯ Low	CRITICAL
**Miscarriage rate**												
3	Randomized trials	Not serious	Not serious	Not serious	Very serious ^d^	None	7/135 (5.2%)	4/128 (3.1%)	**RR 1.71** (0.54 to 5.48)	**22 more per 1000** (from 14 fewer to 140 more)	⨁⨁◯◯ Low	CRITICAL

^a^: optimal information size is not met, and the 95% CI of the RR included RR of 1.25; ^b^: high heterogeneity across studies; ^c^: optimal information size is not met, wide range of 95% CI and the 95% CI of the RR included RR of 1.25; ^d^: optimal information size is not met, wide range of 95% CI and the 95% CI of the RR included RR of 0.75 and 1.25; CI: confidence interval; RR: risk ratio.

**Table 4 bioengineering-10-00303-t004:** Summary of the quality assessment by GRADE approach of outcomes included in the meta-analysis of thin endometrium patients.

Certainty Assessment							No. of Patients		Effect		Certainty	Importance
No. of Studies	Study Design	Risk of Bias	Inconsistency	Indirectness	Imprecision	Other Considerations	PRP	Conventional Treatment	Relative (95% CI)	Absolute (95% CI)		
**Biochemical pregnancy rate**												
1	Randomized trials	Serious ^a^	Not serious	Not serious	Very serious ^b^	None	26/70 (37.1%)	10/73 (13.7%)	**RR 1.97** (1.57 to 2.48)	**133 more per 1000** (from 78 more to 203 more)	⨁⨁◯◯ Low	CRITICAL
**Clinical pregnancy rate**												
1	Randomized trials	Serious ^a^	Not serious	Not serious	Very serious ^b^	Strong association	23/70 (32.9%)	7/73 (9.6%)	**RR 3.46** (1.58 to 7.59)	**236 more per 1000** (from 56 more to 632 more)	⨁⨁◯◯ Low	CRITICAL
**Live birth rate**												
1	Randomized trials	Serious ^a^	Not serious	Not serious	Very serious ^b^	None	11/40 (27.5%)	6/43 (14.0%)	**RR 1.97** (0.80 to 4.83)	**135 more per 1000** (from 28 fewer to 534 more)	⨁◯◯◯ Very low	CRITICAL
**Miscarriage rate**												
1	Randomized trials	Serious ^a^	Not serious	Not serious	Very serious ^c^	None	3/40 (7.5%)	2/43 (4.7%)	**RR 1.61** (0.28 to 9.16)	**28 more per 1000** (from 33 fewer to 380 more)	⨁◯◯◯ Very low	CRITICAL

^a^: one study with unclear risk of bias; ^b^: optimal information size is not met and the 95% of the RR included a RR value of 1.25; ^c^: optimal information size is not met, wide range of the 95% CI of the RR and the 95% of the RR included a RR value of 0.75 and 1.25; CI: confidence interval; RR: risk ratio.

## Data Availability

Data will be made available to the editors of the journal for review or query upon request.

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
