# Peer review of "Efficacy of Platelet-Rich Plasma in Women with a History of Embryo Transfer Failure: A Systematic Review and Meta-Analysis with Trial Sequential Analysis"

_bioengineering, 2023, doi:10.3390/bioengineering10030303_

Round 1

Reviewer 1 Report

The study goal is reasonable, and the study design is well-planned.

I have some comments.

1.     The quality of the referenced studies does not look good because their impact factors were relatively low, which may cause a bias in the analysis.

2.     A lot of references were duplicated in the list. It should be corrected.

3.     It is hard to find a difference between recently published review articles and this manuscript.

Author Response

Reviewer 1:

The study goal is reasonable, and the study design is well-planned.

 I have some comments.

  1. The quality of the referenced studies does not look good because their impact factors were relatively low, which may cause a bias in the analysis.

 Authors: We thank the reviewer for this comment. The risk of bias assessment in the context of systematic review are guided by guidelines. In this study, the methodologic quality of each trial was investigated by using risk of bias assessment in accordance with the Cochrane Handbook for Systematic Reviews of Interventions. (Higgins JPT, 2011a). Each trial was classified at high, low or unclear risk of bias across six specific domains (random sequence generation, allocation concealment, blinding of participants and personnel, blinding of outcome assessment, incomplete outcome data, selective reporting, and other bias). Two authors independently assessed the selected studies.

Using the Cochrane Risk of Bias Tool, each study was classified at high, moderate or low risk of bias. An overall low-risk rating was assigned when none of the six domains were found to be at high risk and if three or less domains were found to be at unclear risk. An overall moderate-risk rating was assigned when one domain was found to be at high risk; or no domains were found to be a high risk but four or more were found to be at unclear risk. In all remaining cases, the trial was classified as having an overall high risk of bias.

  1. A lot of references were duplicated in the list. It should be corrected.

Authors: Sorry for the inconvenience. We have solved this error.

  1. It is hard to find a difference between recently published review articles and this manuscript.

 This is the first systematic review that include TSA analysis to address information size requirement for each outcome. Among the strengths of this systematic review, we can mention the restriction to RCTs to avoid biases from observational studies and the evaluation of the strength of evidence with Grade approach. In addition, most of the studies used similar outcomes definitions and the follow-up time was comparable among studies, avoiding biases caused by the different methodology or trial design.

Reviewer 2 Report

The present systematic review titled “Efficacy of platelet-rich plasma in women with a history of embryo transfer failure: a systematic review and meta-analysis with trial sequential analysis” is of great interest to specialists in the field of reproduction, as it describes in detail the available literature concerning the ability of PRP to manage infertility problems in women with a history of embryo transfer failure. Despite the benefits of PRP demonstrating the ability to enhance clinical and biochemical pregnancy rates, the authors of the present manuscript raise the question of the correct preparation of the PRP, in particular leukocyte and platelet counts,  as these factors might have a clear impact on the measured clinical outcomes.  The authors underline that future studies should elucidate the safety profile of intra-uterine PRP administration. The manuscript contains comprehensive information. The only drawback of this manuscript is the numerous punctuation errors.

Author Response

Reviewer 2:

The present systematic review titled “Efficacy of platelet-rich plasma in women with a history of embryo transfer failure: a systematic review and meta-analysis with trial sequential analysis” is of great interest to specialists in the field of reproduction, as it describes in detail the available literature concerning the ability of PRP to manage infertility problems in women with a history of embryo transfer failure. Despite the benefits of PRP demonstrating the ability to enhance clinical and biochemical pregnancy rates, the authors of the present manuscript raise the question of the correct preparation of the PRP, in particular leukocyte and platelet counts, as these factors might have a clear impact on the measured clinical outcomes.  The authors underline that future studies should elucidate the safety profile of intra-uterine PRP administration. The manuscript contains comprehensive information. The only drawback of this manuscript is the numerous punctuation errors.

Authors: Thank you very much for your comments. We have carefully read the entire manuscript to correct punctuation errors.

Reviewer 3 Report

Dear authors,

here you can find my comments:

1) please correct some typos error. E.G. points before citations.

2) please include some definitions of RIF (ASRM, ESHRE  etc).

4) there are two main serious concerns. The TE has not been reported on outcomes of interest. Please state the "history of failed embryo transfer". It is too generic to obtain proper conclusions. 

5) The GRADE analysis showed CRITICAL outcomes which, accordingly, could exempt authors to perform new studies since it is clear that PRP is efficacy. This is quite doubtable since the risk of bias showed some issues of interest.

Author Response

Reviewer 3:

Dear authors,

here you can find my comments:

  • please correct some typos error. E.G. points before citations.

Authors: Thanks for your remark. We have corrected the errors.

2) please include some definitions of RIF (ASRM, ESHRE  etc).

Authors: In the text we have defined RIF and included a new sentence. We have also insert new references.

4) there are two main serious concerns. The TE has not been reported on outcomes of interest. Please state the "history of failed embryo transfer". It is too generic to obtain proper conclusions.

Authors: The outcomes of PRP use in TE have been reported in Figures 3 and 4 and Table 4. These data were also commented in the text. The review protocol was published in Prospero database (CRD42021287963) where the study population/participant has been the following “This study will include information about women with a history of failed embryo transfer”. This is what has been described in the focus question within the manuscript.

5) The GRADE analysis showed CRITICAL outcomes which, accordingly, could exempt authors to perform new studies since it is clear that PRP is efficacy. This is quite doubtable since the risk of bias showed some issues of interest.

Authors: We thank the reviewer for this comment. In the GRADE analysis the importance of a variable is grade according to their importance in developing guidelines or conclusions regarding the implementation of a treatment/technology. More studies are needed according to the results of this systematic review. Based on our meta-analysis and the subsequent TSA evaluation, autologous PRP has the ability to enhance clinical and biochemical pregnancy rates and the live birth rate. Additional well-designed trials are needed to confirm the beneficial effect of PRP over the implantation rate. Accordingly, there is lack of evidence to draw conclusions on whether PRP can bring benefit on women with TE due to the limited number of studies. Future studies should also elucidate the safety profile of intra-uterine PRP administration with long-term follow-ups. Furthermore, most of available trials were performed in the same country (Iran), being the ethnicity and important confounding factor that can lead to a data misinterpretation. Moreover, further studies are needed to assess the effect of PRP composition and protocol of preparation on the outcomes.

Reviewer 4 Report

The present study aims to investigate the efficacy of platelet-rich plasma (PRP) in women with a history of recurrent implantation failure. The study has been properly conducted and the manuscript has been well written. The use of PRP in assisted reproduction is a matter of debate and the findings of this meta-analysis would contribute to the literature. The methodology of this meta-analysis and, thus, its power is limited by the numerical and geographical restriction for the relevant studies. However, the references have been well chosen so that they are all relevant and up-to-date references. Similarly, all tables and figures are relavant and they have been set up carefully. Therefore, I recommend that the manuscript in its current version can be accepted for publication in Bioengineering.

Author Response

Reviewer 4:

The present study aims to investigate the efficacy of platelet-rich plasma (PRP) in women with a history of recurrent implantation failure. The study has been properly conducted and the manuscript has been well written. The use of PRP in assisted reproduction is a matter of debate and the findings of this meta-analysis would contribute to the literature. The methodology of this meta-analysis and, thus, its power is limited by the numerical and geographical restriction for the relevant studies. However, the references have been well chosen so that they are all relevant and up-to-date references. Similarly, all tables and figures are relevant and they have been set up carefully. Therefore, I recommend that the manuscript in its current version can be accepted for publication in Bioengineering.

Authors: Thank you very much for your recommendation.

Round 2

Reviewer 1 Report

The manuscript is improved by revision. However, a couple of issues remained.

1. In FIGURE 3 and Figure 4, a high-risk bias group is included. Excluding this high-bias group, please submit the meta-analysis again as a separate figure.

2. Reference 44 is a study on the RPL group, which had fetal growth problems, but not implantation problems. Therefore, it seems appropriate to exclude them from this study. In addition, IVF for RPL patients is difficult to see as a logical treatment.

Author Response

Dear editor, here I attach the answer to reviewer 1. We hopefully the article is now suitable for publication. Kind regards.
